# Novel Liquid Biomarker Panels for A Very Early Response Capturing of NSCLC Therapies in Advanced Stages

**DOI:** 10.3390/cancers12040954

**Published:** 2020-04-12

**Authors:** Florian Janke, Farastuk Bozorgmehr, Sabine Wrenger, Steffen Dietz, Claus P. Heussel, Gudula Heussel, Carlos F. Silva, Stephan Rheinheimer, Manuel Feisst, Michael Thomas, Heiko Golpon, Andreas Günther, Holger Sültmann, Thomas Muley, Sabina Janciauskiene, Michael Meister, Marc A. Schneider

**Affiliations:** 1Division Cancer Genome Research, German Cancer Research Center (DKFZ), INF 280, 69120 Heidelberg, Germany; 2German Cancer Consortium (DKTK), INF 280, 69120 Heidelberg, Germany; 3Translational Lung Research Center Heidelberg (TLRC), Member of the German Center for Lung Research (DZL), INF 156, 69120 Heidelberg, Germany; 4Medical Faculty, Heidelberg University, 69117 Heidelberg, Germany; 5Department of Thoracic Oncology, Thoraxklinik at University Hospital Heidelberg, Roentgenstrasse 1, 69126 Heidelberg, Germany; 6Department of Respiratory Medicine, Hannover Medical School, Carl-Neuberg-Str. 1, 30625 Hannover, Germany; 7Biomedical Research in Endstage and Obstructive Lung Disease Hannover (BREATH), Member of the German Center for Lung Research (DZL), Carl-Neuberg-Str. 1, 30625 Hannover, Germany; 8Department of Diagnostic and Interventional Radiology with Nuclear Medicine, Thoraxklinik at University Hospital Heidelberg, Roentgenstrasse 1, 69126 Heidelberg, Germany; 9Department of Diagnostic and Interventional Radiology, University Hospital Heidelberg, INF 400, 69120 Heidelberg, Germany; 10Department of Pneumology, Thoraxklinik at University Hospital Heidelberg, Roentgenstrasse 1, 69126 Heidelberg, Germany; 11Institute of Medical Biometry and Informatics (IMBI), University of Heidelberg, Im Neuenheimer Feld 130.3, 69120 Heidelberg, Germany; 12Department of Pneumology, Hannover Medical School, Carl-Neuberg-Str. 1, 30625 Hannover, Germany; 13Cardio-Pulmonary Institute (CPI), Klinikstr. 33, 35392 Giessen, Germany; 14Universities of Giessen and Marburg Lung Center (UGMLC), Member of the German Center for Lung Research (DZL), Klinikstr. 33, 35392 Giessen, Germany; 15Translational Research Unit, Thoraxklinik at University Hospital Heidelberg, Roentgenstrasse 1, 69126 Heidelberg, Germany

**Keywords:** NSCLC, early response biomarkers, liquid biomarkers, targeted therapy, chemotherapy

## Abstract

Computed tomography (CT) scans are the gold standard to measure treatment success of non-small cell lung cancer (NSCLC) therapies. Here, we investigated the very early tumor response of patients receiving chemotherapy or targeted therapies using a panel of already established and explorative liquid biomarkers. Blood samples from 50 patients were taken at baseline and at three early time points after therapy initiation. DNA mutations, a panel of 17 microRNAs, glycodelin, glutathione disulfide, glutathione, soluble caspase-cleaved cytokeratin 18 (M30 antigen), and soluble cytokeratin 18 (M65 antigen) were measured in serum and plasma samples. Baseline and first follow-up CT scans were evaluated and correlated with biomarker data. The detection rate of the individual biomarkers was between 56% and 100%. While only keratin 18 correlated with the tumor load at baseline, we found several individual markers correlating with the tumor response to treatment for each of the three time points of blood draws. A combination of the five best markers at each time point resulted in highly significant marker panels indicating therapeutic response (R^2^ = 0.78, R^2^ = 0.71, and R^2^ = 0.71). Our study demonstrates that an early measurement of biomarkers immediately after therapy start can assess tumor response to treatment and might support an adaptation of treatment to improve patients’ outcome.

## 1. Introduction

Lung cancer is the most common cause of cancer-related death worldwide with non-small cell lung cancer (NSCLC) being the predominant entity with approximately 85% of cases [1]. Treatment options have changed considerably within the past years, especially with the advent of tyrosine-kinase-inhibitors (TKI) for epidermal growth factor receptor (EGFR) mutation and anaplastic lymphoma kinase (*ALK*) rearrangement positive NSCLC, which comprises about 15% and 2% of NSCLC cases, respectively [2,3]. TKI therapies result in substantial survival benefits for the patient [4,5,6]. However, most patients eventually experience relapse as a result of acquired TKI resistance [7,8]. For patients without targetable mutations, platinum-based chemotherapy (±pembrolizumab) is still the treatment of choice, except for patients with programmed death ligand 1 (PDL1) expression >50%, who can receive immunotherapy with pembrolizumab as a monotherapy.

Regardless of the type of therapy, the gold standard for clinical monitoring of therapy efficacy in lung cancer patients is change in tumor size according to Response Evaluation Criteria in Solid Tumors, version 1.1 (RECIST-1.1) criteria. However, these changes are often evident in CT and MRI imaging with a delay of several weeks after initiation of systemic treatment and, therefore, unsuitable for frequent therapy monitoring and early assessment of response. The analysis of circulating biomarkers such as circulating tumor DNA (ctDNA), circulating microRNAs (miRNAs), and disease-associated protein markers offers a minimal-invasive tool to overcome this limitation, permitting frequent sample collection and timely assessment of the patient’s disease status. This would allow for early detection of non-responders and avoid side effects and costs of ineffective treatment.

Preclinical and clinical studies have shown that apoptosis significantly increases 24 h after chemotherapy administration [9,10,11]. These changes in tumor biology are expected to result in changes in plasma/serum levels of (i) defined proteins relating to tumor cell death and regressions and (ii) release of free DNA, RNA carrying mutations and/or gene fusions.

Previous studies have shown that *EGFR* mutations can be detected in the plasma of patients with high prevalence, reflecting the landscape and heterogeneity of primary tumors and metastases in NSCLC [12,13]. Serial evaluation of mutant DNA could provide a noninvasive assessment of therapy response and tumor progression, including the detection of resistance mutations or an increase of EGFR sensitizing mutations associated with clinical progression.

MicroRNAs are short noncoding RNA molecules known as important regulators of gene expression. Deregulation of miRNAs is frequently observed in human cancers, including lung cancer, and is considered one of the characteristics of malignant transformation [14]. With their high stability, circulating miRNAs can be detected robustly in plasma and, therefore, represent promising biomarkers in cancer patients [15,16].

Another sign of tumor cells undergoing apoptosis is the increase of caspase-cleaved cytokeratin 18 fragments (M30 antigen). In the circulation, CK-18 occurs as a full-length protein (M65 antigen) as well as the 21-kDa caspase-cleaved fragment if epithelial cells undergo apoptosis [17]. Previous studies have demonstrated that serum levels of CK-18 proteins can be useful as an independent factor in predicting response to chemotherapy in patients with NSCLC [18].

Glutathione (GSH) is a tripeptide of glutamate, cysteine, and glycine, a potent antioxidant found at high concentration in all tissues. Under normal conditions, the majority of GSH exists in reduced form (0.5 to 10 mM). However, when GSH interacts with free radicals or acts as a cofactor for antioxidant enzymes, such as GSH peroxidases, oxidized glutathione (oxGSH) is generated [19]. Increased glutathione levels and glutathione-S-transferase activity have been implicated in platinum neutralization and resistance. The correlation between increased glutathione levels and drug resistance has been documented in a variety of tumors [20].

A further target used in this study was the glycoprotein glycodelin, which has been well characterized in females [21]. It is secreted by the inner layers of the endometrium and highly expressed during the first trimester of pregnancy. Glycodelin has been shown to regulate the invasion of the trophoblast into the endometrium and the immunotolerance of the maternal immune system [22]. However, several studies have demonstrated the expression and of glycodelin in hormone-regulated cancers, i.e., ovarian cancer [23] and breast cancer [24], as well as in non-hormone-regulated cancers such as melanoma [25] and lung cancer [26].

This study aimed to define predictive marker panels indicating a successful or failing tumor therapy at very early time points after therapy initiation. Therefore, each of the described biomarkers was evaluated separately and in combination for their potential as predictive therapy markers at very early time points (day +1, day +7, day +14) in patients with advanced NSCLC.

## 2. Results

### 2.1. Biomarker Detection

For this study, we collected serum and plasma from 50 NSCLC patients. Patients were divided into two groups depending on disease treatment. Group A (*n* = 25) received conventional chemotherapy since no targetable molecular alteration was detected during routine diagnostics. Group B (*n* = 25) consisted of patients with a driver mutation or gene fusion targetable with a tyrosine kinase inhibitor (see patients’ characteristics, Table 1). First blood sample was collected within 24 h prior to therapy start (day −1, Figure 1). Due to different therapy concepts, blood samples were collected at different time points after therapy start. For the chemotherapy group (group A), one post-treatment sample was collected at day +1, while two blood samples were assembled at day +7 and +14 for the TKI group (group B). In the first approach, patients’ tumor response to therapy was assessed (Figure 2A). The TKI patients (group B) showed better therapy response compared to patients treated with chemotherapy. Using RECIST-1.1 criteria, 21 of the 50 patients revealed a response to therapy with a partial remission. For two patients, progress of the disease was diagnosed at the time of follow-up.

Fifteen patients had a slightly lower tumor load, six patients exhibited a stable tumor load, and four patients exhibited a slight progression of the tumor diagnosed as stable disease. The investigated biomarkers (Table 2 and Appendix A
Table A1) were detected in 100% of the patient groups, except glycodelin (60%) and the driver mutation of the TKI group (56%) (Figure 2B). Quality control for mutation detection and RNA measurements revealed that the investigated plasma samples had no or only a low risk to be affected by hemolysis (Table A2). For two patients from group A, no CT data were available, and for another patient from group B, miRNA analyses and mutation detection failed due to low amount of blood sample. These three patients were excluded from the subsequent analyses.

### 2.2. Correlation of Tumor Load and Biomarker Detection at Time Point of Therapy Start

First, we were interested in whether the serum/plasma levels of the selected biomarkers correlate with tumor load at the time of diagnosis (Figure 3A). The tumor load was assessed by an experienced radiologist and defined as the size of the tumor(s) at the time point of therapy start. Afterward, the different biomarkers were determined and correlated to tumor load using Pearson correlation coefficient. Interestingly, the plasma levels of biomarkers M30 and M65 (cleaved and full-length cytokeratin 18) correlated significantly with the tumor load of the patients (r = 0.55, *p* < 0.001 and r = 0.42, *p* = 0.002, Figure 3B). All other markers failed to correlate in our cohort at the time of diagnosis.

### 2.3. Correlation Analyses of Single Biomarkers and Tumor Load Change

To investigate whether each of the biomarkers can individually indicate an early response to therapy, we first correlated every single marker with the change of tumor load. To do so, we examined the predictive value of the single markers in serum and plasma at the three time points using a linear regression model comprising the variable “relative tumor load change from baseline to first CT after therapy” (Figure 4). For group A, the microRNA *hsa-miR-210-3p* significantly correlated with the tumor load change (r = 0.49, *p* = 0.017), while for group B (day +7) *hsa-miR-134-5p* showed the best correlation with the tumor load change (r = −0.47, *p* = 0.020). *hsa-miR-23a-3p* and *hsa-miR-134-5p* also correlated partly (r = −0.43, *p* = 0.036 and r = −0.41, *p* = 0.049). For group B, the mutation detected in cell-free DNA (cfDNA) correlated best (r = 0.61, *p* = 0.020, Figure 4). Using Pearson correlation coefficient, heatmaps combining every single marker also revealed that there are highly correlated biomarkers at every time point, for example, M65 and M30.

### 2.4. Stepwise Regression Model

Many different markers have been described to be useful individually for diagnosis or prediction of NSCLC therapies [36]. However, due to complex biological changes in tumor and patients’ metabolism, no single biomarker has entered clinical routine. Hypothesizing that a marker panel might be superior to single markers, we performed a five-step forward regression analysis (based on the Akaike Information Criterion) to find a marker panel consisting of the 23 investigated biomarkers with the best predictive performance for each time point (Figure 5). Using marker panels, the correlation of the tumor load change and the marker abundance change increased to R^2^ = 0.78 (group A, time point +1), R^2^ = 0.71 (group B, time point +7), and R^2^ = 0.71 (group B, time point +14) (Figure 5A). Thereby, *hsa-miR-20a-5p* was the only marker included in more than one panel. Interestingly, none of the individual biomarker panels was predictive at the other investigated time points (Table A3). As an example, while the marker panel for group B at day +7 correlated well with the tumor load change (R^2^ = 0.71), a low correlation was observed for group A at day +1 (R^2^ = 0.06) and for group B at day +14 (R^2^ = 0.25). Visualization of each panel showing the marker abundance change in relation to tumor load change confirmed the strengths of the individual panels (Figure 5 B).

Tumor load change for each patient from baseline to follow-up was indicated in green (reduced tumor load), grey (no tumor load change), and red (increased tumor load change). While all patients with a lower marker panel value also had a reduced tumor load (green lines), only patients with no (grey lines) or increased (red lines) tumor load change showed an increased marker panel abundance change. Using the three best markers instead of the five markers panel resulted in a decrease of predictive robustness (R^2^ = 0.62 for group A, time point +1; R^2^ = 0.61 for group B, time point +7, and R^2^ = 0.54 for group B, time point +14).

## 3. Discussion

The field of biomarkers in NSCLC is very diverse with a variety of methods and biomarkers investigated in the past. Besides classical tissue-based markers that have been used for decades [37], modern methods analyzing proteomes, epigenomes, or metabolomes enlarged the generally available biomarker resources [38]. Liquid biopsies have attracted increasing attention in recent years for disease diagnosis, prognosis, and treatment, especially in late stages and with poor general condition of the patients [39]. An early assessment of liquid biomarkers has demonstrated that changes in circulating tumor DNA (ctDNA) or microRNA levels might indicate therapy response [40,41,42].

The current study was conducted to investigate whether a very early assessment of different blood-based biomarkers might indicate a tumor response to therapy. We selected 23 biomarkers that have been shown to be relevant in liquid biopsy approaches, including prognosis or therapy response. These markers were measured in two patient groups (group A, *n* = 25, chemotherapy cohort and group B, *n* = 25, TKI cohort). Considerable hemolysis, which might influence the quality and composition of the investigated microRNAs [43], was not observed in the plasma samples of our cohort. Glycodelin, which has been described as a prognostic liquid biomarker in NSCLC, was detectable in only 60% of all samples. This is in line with literature data where it was demonstrated that glycodelin is measurable only in particular serum samples [27]. Similar observations were made for ctDNA. Here, the detection rate in group B was only 56%. This might be explained by the fact that the ctDNA is markedly diluted by circulating germline DNA. Similar rates have been published investigating EGFR resistance mutations in liquid biopsies [35,44,45]. Interestingly, nearly all investigated markers failed to correlate with the tumor load at baseline. Contrary to our expectations, although the copy number of mutations and the glycodelin serum concentrations correlated within group B at time point +7, the biomarkers glycodelin and the driver mutation abundance did not correlate with the tumor size. We expected a correlation since both markers are exclusively expressed and released by malignant cells. Only the apoptosis markers M65 and M30 (full-length and cleaved cytokeratin 18) correlated with tumor load and might be useful markers for diagnosis, which confirms published data [46].

The analysis of biomarker abundance and change of tumor load revealed that for group A (day +1) and group B (day +7) only one microRNA each (*miR-210-3p* and *miR-134-5p*) was significant. This is in line with published data since *miR-210-3p* has been described to be a marker decreasing after chemotherapy treatment of patients with NSCLC, while *miR-134-5p* is upregulated in gefitinib resistance cell lines. For day +14 (group B), three out of the 23 biomarkers correlated significantly with the tumor load change. Here, the mutational status was the best marker at this time point. In addition, *miR-23a-3p* and *miR-134-5p* were also found to be significant predictors for tumor response to treatment at day +14 after therapy initiation. Since our cohort size was limited and our goal was to define a marker panel for valid evidence, we performed a stepwise forward regression model with the five most promising biomarkers for each time point. Indeed, we reached a very high model fit for all three measured groups (R^2^ = 0.78 (day +1), R^2^ = 0.71 (day +7 and +14)) considering the relatively small cohort size. Due to this statistical method and the fact that glycodelin as well as the driver mutation were not detectable in all patients, the number of patients included in the analyses dropped down from 25 to *n* = 16 (day +1), *n* = 19 (day +7), and *n* = 15 (day +14) in the three groups. Using a panel consisting of five markers in this relatively small cohort, therefore, might indicate an overfitted model. However, a reduction to a three-marker based panel still reached high predictive values at the three time points (R^2^ = 0.62 (day +1), R^2^ = 0.61 (day +7), and R^2^ = 0.54 (day +14)). We also observed that the marker panels cannot be transferred to the other time points of blood sample collection. On the one hand, this might result from different cellular processes since the time between the three blood assessments was approximately one week each. The complex interaction of cellular therapy response, as well as apoptotic processes, probably influenced the amounts of RNA, DNA, and proteins released into the blood. On the other hand, the small cohort size for both groups might also be an explanation for these observations. Therefore, our findings have to be validated in a larger cohort.

Currently, patients need to receive a tumor-specific therapy at least 4 to 6 weeks, before conventional radiographic tumor assessments can reliably determine a tumor response. From a clinical perspective, identification of non-responders using a marker or marker panel that is able to indicate success or failure of a therapy at an early stage is highly warranted. The use of such marker panels might help to adjust treatment earlier than currently done using radiographic tumor assessments and, thereby, avoid or shorten side effects and adverse events of ineffective treatment. Consequently, early detection of therapy success would lead to a better quality of life especially for patients in late stages of tumor disease. Furthermore, an early adaption of therapy concept might impact the progression-free or overall survival of patients. To our knowledge, except for the detection of mutations, no studies have used these very early time points for prediction of therapy response in NSCLC before.

## 4. Material and Methods

### 4.1. Study Design and Patient Biospecimen

The early response biomarker study was a part of the early response trial (ERT), a multicenter study funded by the German Center for Lung Research (DZL). This biomarker study was conducted in accordance with the Reporting Recommendations for Tumor Marker Prognostic Studies (REMARK) [47]. Serum and plasma samples were provided by Lung Biobank Heidelberg, Hannover Unified Biobank HUB, and UGMLC-Giessen Biobank, members of the Biobank platform of the German Center for Lung Research (DZL). The use of biomaterial and data for this study was approved by the local ethics committee of the Medical Faculty Heidelberg (S-445/2015). All patients included in the study signed informed consent and the study performed according to the principles set out in the WMA Declaration of Helsinki. We investigated blood from 50 stage IV patients with NSCLC (Table 1). Patients were selected by clinicians requiring MRI and measurable tumor size of at least 2 cm. All patients suffered from NSCLC and were selected by recommended therapy concept according to the current guidelines. Group A (*n* = 25) received conventional chemotherapy since no targetable molecular alteration was detected during routine diagnostics. Group B (*n* = 25) contained patients with a targetable driver mutation or gene fusion (see patients’ characteristics, Table 1). First blood sample was collected within 24 h prior to therapy start (day −1, Figure 1). Due to different therapy concepts, blood samples were collected at different time points after therapy start. For chemotherapy group (group A), one post-treatment sample was collected at day +1, while two blood samples were assembled at day +7 and +14 for the TKI group (group B, Figure 1). Time points were selected due to literature search and own observations [35] that intravenous uptake of drugs can be faster compared to oral medication [48]. Routine computed tomography (CT) was performed at baseline and at time point of first clinical restaging (in median after 50 days). Tumor load change (primary and metastasized sites) was assessed by an experienced radiologist according to RECIST v 1.1 as size change of the tumor at time point of therapy start and at restaging.

### 4.2. DNA Extraction and Analysis by Digital PCR

cfDNA was extracted from one mL aliquots of frozen plasma with the QIAamp circulating nucleic acid kit (Qiagen, Hilden, Germany), following the manufacturer’s recommendations for the purification of circulating nucleic acids. DNA quality was assessed with the Bioanalyzer 2100 using the High Sensitivity DNA Kit (Agilent Technologies, Santa Clara, CA, USA). DNA was quantified using the Qubit dsDNA HS Assay Kit with the Qubit 2.0 fluorometer (Thermo Fisher Scientific, Waltham, MA, USA). cfDNA was subjected to the measurement of known sensitizing EGFR mutations by digital PCR (dPCR) and TaqMan liquid biopsy dPCR assays EGFR_6223, EGFR_6224 and EGFR_6225 (Thermo Fisher Scientific, Waltham, MA, USA), following the manufacturer’s protocol. For echinoderm microtubule associated protein-like 4 (EML4)-ALK fusion and Serine/threonine-protein kinase B-raf (rapidly accelerated fibrosarcoma) (BRAF) mutation, no assay was available. Samples were measured as technical triplicates using the QuantStudio 3D digital PCR instrument and subsequently analyzed with the QuantStudio 3D AnalysisSuite Software (Thermo Fisher Scientific, Waltham, MA, USA). Reported mutated copies per µL reaction volume were extrapolated to mutant copies/mL plasma.

### 4.3. miRNA Extraction and Analysis by Quantitative Reverse-Transcription PCR

For this study, we selected 17 miRNAs that have been described in literature to be diagnostic and/or prognostic/predictive indicators when measured from plasma/serum of patients (Table A1). Circulating miRNAs were extracted from one milliliter aliquots of frozen plasma. Each plasma sample was spiked with 4.8E+08 copies of *cel-miR-54* to assess isolation efficiency. The isolation was performed using the QIAamp circulating nucleic acid kit (Qiagen, Hilden, Germany), following the manufacturer´s recommendations for the purification of circulating miRNAs. Isolated miRNAs were reverse transcribed (RT) with the TaqMan microRNA reverse transcription kit and reverse-transcription primers of the TaqMan microRNA assay kits for miRNAs *hsa-miR-21-5p*, *hsa-miR-214-3p*, *hsa-miR-23a-3p*, *hsa-miR-221-3p*, *hsa-miR-222-3p*, *hsa-miR-134-5p*, *hsa-miR-126-3p*, *hsa-miR-103-3p*, *let-7e-5p*, *hsa-miR342-3p*, *hsa-miR-1290*, *hsa-miR-223-3p*, *hsa-miR-20a-5p*, *hsa-miR-145-5p*, *hsa-miR-628-3p*, *hsa-miR-29c-3p*, *hsa-miR-210-3p*, *has-miR191*, and *hsa-miR-451a* (Thermo Fisher Scientific, Waltham, MA, USA). The qPCR reaction was conducted as technical triplicates in 384-well plates with the RT product, Premix Ex Taq master mix (Takara Bio, Kusatsu, Japan) and TaqMan probes for each miRNA using the LightCycler480 instrument (Roche Diagnostics, Basel, Switzerland). Reverse transcription and qPCR were performed following the manufacturer´s protocol. miRNA abundances were normalized to the abundance of *hsa-miR-191*, which has been shown to be highly detectable in plasma and recommended to be used as housekeeper miRNA [43,49], within each sample.

### 4.4. Measurement of Cytokeratin 18 (M30 and M65 Determination)

M30 and M65 levels in Ethylenediaminetetraacetic acid (EDTA)-Plasma were determined with M30 Apoptosense and M65 ELISA kits (PEVIVA, Tecomedical, Buende, Germany) according to the manufacturer’s protocol. The samples were undiluted or diluted by a factor 2. The assay range was 0–1000 U/L for M30 and 0–2000 U/L for M65. For M30, units were defined using a recombinant protein standard. For M65, the units were defined using a synthetic peptide containing M6 and M5 epitope. 1 U/L = 1.24 pM.

### 4.5. Determination of GSH and oxGSH Concentrations

GSH and oxGSH were detected with the competitive EIA kits “All species Glutathione ELISA kit” and “Human Oxidized Glutathione ELISA kit” from LSBio (Seattle, Washington, USA) according to the manufacturer’s protocol. EDTA-plasma samples were diluted by a factor of 2 with sample diluent. For analysis, the data were linearized by plotting them on logarithmic axes. The detection range was 1.23–100 µg/mL with a sensitivity of 0.45 µg/mL for GSH and 4.688–300 pg/mL with a sensitivity of 2.8 pg/mL for oxGSH.

### 4.6. Measurement of Glycodelin

Glycodelin levels in sera were detected using an enzyme-linked immunosorbent assay kit (ELISA BS-30-20, Bioserv Diagnostics, Rostock, Germany) with 50 µL of each serum in two technical replicates. The readout and standard curve were performed with ELISA Reader (Tecan Group Ltd., Crailsheim, Germany). ELISA results were visualized with GraphPad Prism 5 (GraphPad Software, San Diego, CA, USA).

### 4.7. Statistical Analyses

The patient cohort was described using median and range for continuous variables and using absolute and relative frequencies for categorical variables. Furthermore, tumor response for each patient (in % RECIST) was descriptively illustrated by a waterfall plot. Afterward, correlations between individual biomarkers and tumor load at baseline were assessed using Pearson correlation coefficient. Examining the predictive value of single markers at the three time points, a univariate linear regression model was build comprising the variable “relative tumor load change from baseline to first CT after therapy” as a dependent variable for each biomarker and time point. Furthermore, for each time point, correlations between the biomarker were assessed using Pearson correlation coefficient and were illustrated by heatmaps. To create marker panels for the prognosis of relative tumor load change, a five-step forward regression analysis (based on the Akaike Information Criterion, AIC) was performed to find the marker panel consisting of biomarkers with the best predictive performance for each point in time. Due to the exploratory character of the study, *p*-values have a descriptive meaning and are not adjusted for multiplicity. *p*-values <0.05 are defined as statistically significant. Furthermore, no missing values were imputed. All analyses were performed using R version 3.5.1 (https://www.r-project.org) and Graph Pad Prism Version 5 (GraphPad Software, San Diego, CA, USA).

## 5. Conclusions

For the first time, we investigated a panel of liquid biomarkers at three very early time points after therapy initiation (day +1, day +7, and day +14) for their predictive value. We found three individual marker panels including five biomarkers each at every time point with correlation rates R^2^ > 0.71 to tumor load change. These marker panels highly implicate that the efficiency of a specific NSCLC therapy can already be measured at very early time points after therapy start and might help to avoid or shorten side effects and adverse events of ineffective treatment.

## Figures and Tables

**Figure 1 cancers-12-00954-f001:**
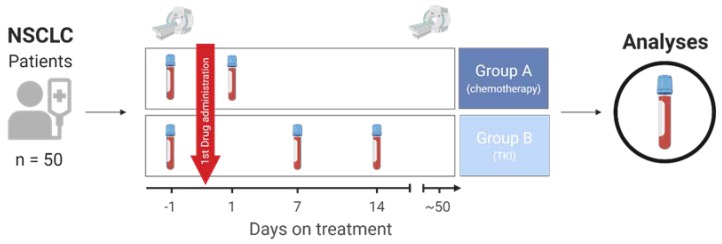
Description of the study concept. Blood from 50 patients with non-small cell lung cancer (NSCLC) was collected at baseline one day prior to therapy start (day −1) and after therapy initiation (day +1 for group A, day +7 and +14 for group B). Routine computer tomography (CT) at baseline and at time point of first clinical restaging was evaluated for tumor load change. Restaging CT was assessed in median at day 50. TKI: tyrosine kinase inhibitor.

**Figure 2 cancers-12-00954-f002:**
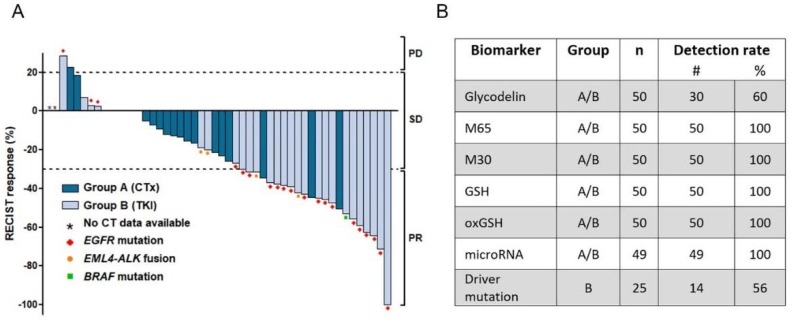
Patient response and detection rates of biomarkers. (**A**) Waterfall plot of tumor response to therapy. Tumor load was evaluated by an experienced radiologist using Response Evaluation Criteria in Solid Tumors, version 1.1 (RECIST-1.1) criteria. Dotted lines indicate thresholds for definition of progressive disease (PD), stable disease (SD), or partial remission (PR). Group A included patients treated with platinum-based chemotherapy, group B consists of patients receiving targeted therapy. (**B**) Detection efficiency of the biomarkers measured in both groups. CTx: Platinum-based chemotherapy, TKI: Tyrosine kinase inhibitor, EGFR: Epidermal Growth Factor Receptor, EML4-ALK: echinoderm microtubule associated protein-like 4-anaplastic lymphoma kinase, BRAF: Serine/threonine-protein kinase B-raf (rapidly accelerated fibrosarcoma), M65: Intact and caspase-cleaved Cytokeratin 18, M30: caspase-cleaved Cytokeratin 18, GSH: Glutathione, oxGSH: Oxidized glutathione.

**Figure 3 cancers-12-00954-f003:**
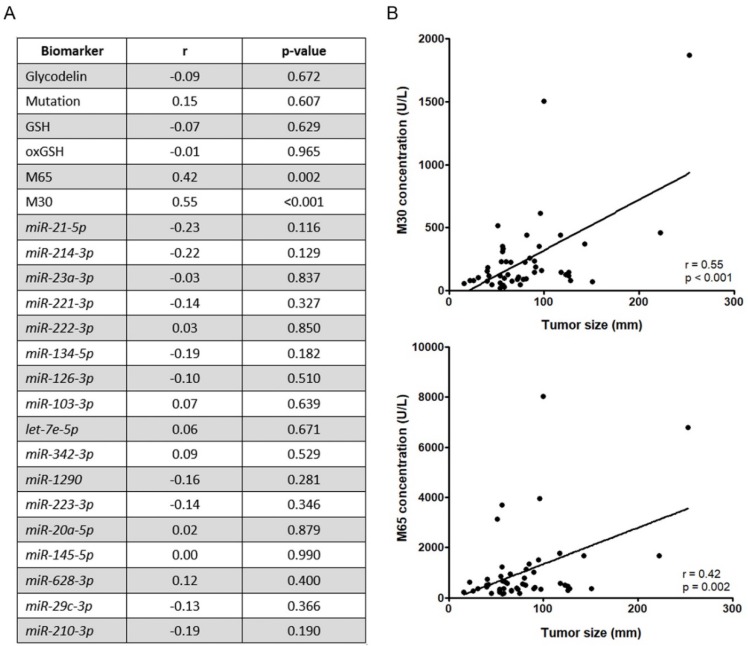
Correlation of markers and tumor load at baseline. (**A**) Linear regression analyses of the individual biomarkers and the tumor load at baseline time point. (**B**) Correlation plots of the two biomarkers with the highest correlation (M30 and M65).

**Figure 4 cancers-12-00954-f004:**
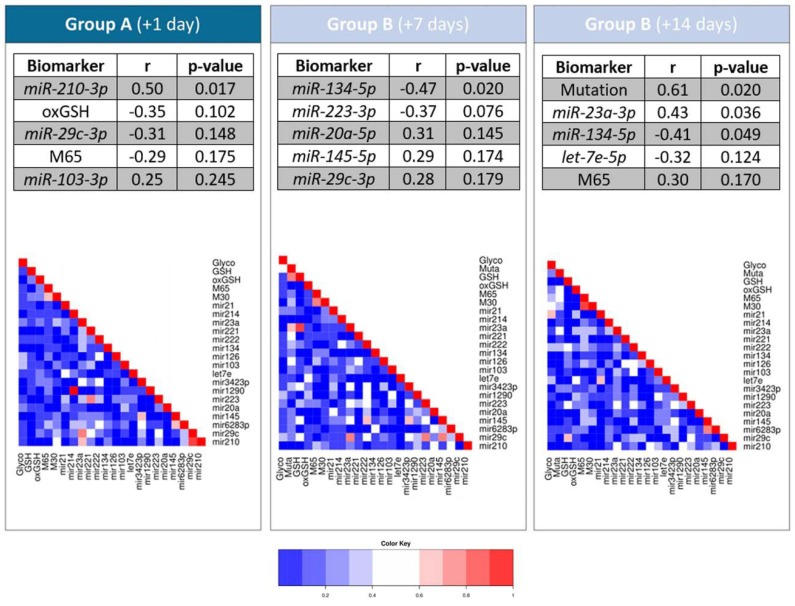
Predictive value of single markers at the three time points and correlation analyses. Linear regression analysis results of the individual markers at the three time points in relation to relative tumor load change from baseline to first CT after therapy. Heatmap indicates correlation between the single biomarkers (Pearson correlation).

**Figure 5 cancers-12-00954-f005:**
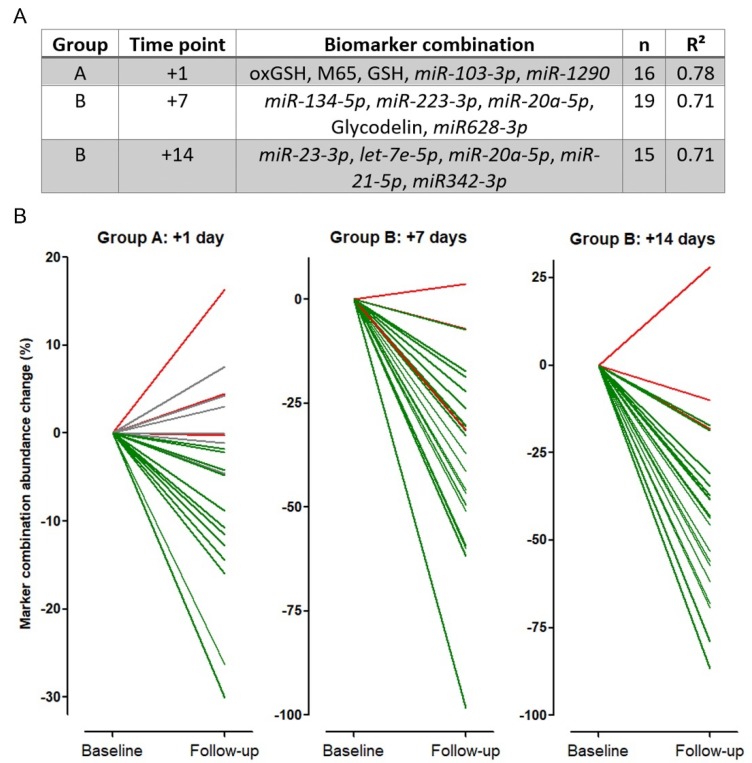
Stepwise regression model and marker combination abundance change. (**A**) Five-step forward regression analysis using the five best markers for each time point. (**B**) Biomarker combination from (**A**) in relation to tumor load change. Every line represents one patient. green = patients with tumor load reduction, red = patients with tumor growth, grey = patients without tumor load change at first CT scan after therapy start.

**Table 1 cancers-12-00954-t001:** Patient characteristics.

Parameter	*n*	(%)
median age in years (range)	62 (40–84)	
gender	50	100
male	24	48
female	26	52
ECOG	50	100
0	25	50
1	21	42
no data	4	8
Smoking status	50	100
current smoker	17	34
ex-smoker <6 months	4	8
ex-smoker >6 months	17	34
non-smoker	10	20
no data	2	4
histology	50	100
non-small cell lung cancer	50	100
adenocarcinoma	46	92
squamous cell carcinoma	1	2
large cell carcinoma	1	2
NOS	2	4
clinical stage (8th edition)	50	100
stage IVA	21	42
stage IVB	29	58
therapy	50	100
chemotherapy *	25	50
targeted therapy	25	50
EGFR **	20	40
EML4-ALK ***	4	8
BRAF	1	2

ECOG: Eastern Cooperative Oncology Group Performance Status Scale, NOS: non other specified, EGFR: Epidermal Growth Factor Receptor, EML4-ALK: echinoderm microtubule associated protein-like 4-anaplastic lymphoma kinase: BRAF: Serine/threonine-protein kinase B-raf (rapidly accelerated fibrosarcoma).* 20 patients received Carboplatin/Pemetrexed, 3 patients Cisplatin/Pemetrexed, 1 patient Carboplatin/nab-Paclitaxel, 1 patient received Cisplatin/Alimta/Avastin. ** 12 patients received afatinib, 6 patients erlotinib, 1 patient gefitinib, 1 patient received nazartinib/capmatinib. *** 2 patients received alectinib, 2 patients received crizotinib. The BRAF patient received trametinib.

**Table 2 cancers-12-00954-t002:** **Biomarkers.** Overview of investigated biomarkers.

Biomarker	Application in Liquid Biopsy	References	Group
Glycodelin	Glycodelin is secreted by non-small cell lung cancer (NSCLC) cells and has predictive value when measured in the serum of patients.	[26,27]	A/B
Cytokeratin-18	Full length (M65) and caspase-cleaved (M30) forms of cytokeratin-18 are increased in lung cancer patients and correlate with apoptosis.	[17,28,29]	A/B
	Glutathione (GSH) and oxidized glutathione (oxGSH) protect cancer cells against cytotoxic compounds and are overexpressed in NSCLC cell lines.	[30,31]	A/B
microRNA	Deregulation of miRNA is associated with various diseases including cancer. Circulating miRNAs show variable abundances in lung cancer patients and healthy individuals, which may be useful for diagnosis, prognosis, and therapy monitoring. An overview of the miRNAs selected for this study is provided in Table A1.	[32,33]	A/B
Driver mutation	Mutations detectable in circulating DNA can reflect the landscape of primary tumors and metastases. Serial evaluation of mutant DNA could provide a noninvasive assessment of therapy response.	[34,35]	B

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
