# Peer review of "Novel Liquid Biomarker Panels for A Very Early Response Capturing of NSCLC Therapies in Advanced Stages"

_cancers, 2020, doi:10.3390/cancers12040954_

Round 1

Reviewer 1 Report

This manuscript investigated the potential usefulness of panel of liquid biomarkers for the assessment of early therapeutic response in advanced NSCLC. The authors measured the presence of several markers (alone and combination) in blood and plasma samples at appropriate time points in therapy, and then performed extensive statistical analyses in an attempt to correlate these data with the tumor load. Although there are no clear correlations, due to the small size of the group (n=25), these results may be valuable for future research on predictive biomarkers.

In my opinion, the presented article should interest Cancers readers. However, the authors must be improved and reorganized few part of text.

Did the authors correlate the results of their research with data on tumor heterogeneity?

Can the authors explain why the biomarker of driver mutation has been tested only for group B?

“Blood samples from patients were taken at baseline and day +1 for the chemotherapy cohort and at baseline, day +7 and +14 for the targeted therapy cohort.” Why is there a difference in sampling time between therapies?

Have the results been validated? How many times have tests been repeated on the samples concerned? In part of the materials and methods there is no information.

Some other comments:

There are several formatting and grammar mistakes. Please carefully revise.

The Authors must be improved the abstract. Also, should add information about miRNA results. Conclusion should be improved and little extended.

The first part of results (biomarker detection) must be rewritten. In current form it is unclear. The signatures of the Figures blend in with the text. Moreover, the authors should add to each correlation an explanation of how they conducted their analyses (move appropriate part from materials and method). Also, there is no information on the type of chemotherapy and TKI-targeted therapies in the cohorts concerned. A reorganization will make the results better to read.

Please provide a list of abbreviation, so that the reader will be easy to follow it.

Author Response

We thank the reviewer for the conctructive comments. The valuable notes have improved the quality of the manuscript. Please find attached our point-by-point comments:

Did the authors correlate the results of their research with data on tumor heterogeneity?

All patients investigated in this study were diagnosed at stage IV. Tumor histology was assessed taking small biopsies from primary tumor or a (distant) metastasis. Since they did not undergo surgery, a tumor broader heterogeneity was not described by the pathologists and cannot be correlated to our results.

Can the authors explain why the biomarker of driver mutation has been tested only for group B?

All patients investigated in this study were tested for the most common molecular alterations in NSCLC (approx. 40). This testing included a variety of driver mutations (EGFR, KRAS, etc.) and fusions (ALK, RET, etc.). All patients without any targetable mutation/fusion (n = 25) received a classical chemotherapy (group A) while patients with EGFR/BRAF mutation or ALK fusion received a tyrosine kinase inhibitor therapy (group B). Therefore, the detection of driver mutations in group A would not make sense.

We are thankful for this hint and added more information to the material and method (MM) (lines 310-320) and result (lines 122-130) section to make this point clearer.

“Blood samples from patients were taken at baseline and day +1 for the chemotherapy cohort and at baseline, day +7 and +14 for the targeted therapy cohort.” Why is there a difference in sampling time between therapies?

The study design is a result of our literature search. There are hints that a classical chemotherapy can affect the tumor faster than a TKI treatment, which is classically administered orally compared to administration of intravenous chemotherapy (see also introduction part). In contrast to intravenous chemotherapy which diffuses into the organs and thereby into tumor tissue very shortly after i.v. administration, oral TKIs need between 7 to 14 days to reach a steady state in the patient’s system. Thus, we examined patients under TKI with a different chronology as compared to the patients undergoing chemotherapy. We added more information on this to the MM section (lines 310-324).

Have the results been validated? How many times have tests been repeated on the samples concerned? In part of the materials and methods there is no information.

The question is not clear. If validated means “technical repeats”, we already added the information in the MM section. In principle, a repetition of the measurements in a different cohort as “validation” has not been done so far since we would have to collect blood from more patients to do this. We broadly discussed the fact that the cohort size of 25 patients for each group needs a future validation.

There are several formatting and grammar mistakes. Please carefully revise.

We apologize for the formatting and grammar mistakes. We carefully revised the text to improve the manuscript.

The Authors must be improved the abstract. Also, should add information about miRNA results. Conclusion should be improved and little extended.

We adapted the abstract to give a better overview of the study (lines 42-55). In our opinion, the results part already describe and includes the miRNA results. Nevertheless, we pointed out in the “stepwise regression model” part that one panel exclusively included miRNAs (line 217).

The first part of results (biomarker detection) must be rewritten. In current form it is unclear.

We agree and reorganized the first section of the results part and added more information about the differences of the groups and the time points of blood collection. We believe that this part is now much clearer and easier to follow (lines 122-133).

The signatures of the Figures blend in with the text.

This expression and the question behind is not clear for us. Maybe it is a formatting problem during PDF production of manuscript for the review process. When the final manuscript is arranged, there should be no more blend visible. Moreover, we added a list of abbreviation in the appendix section and therefore reduced the text below the figures.

Moreover, the authors should add to each correlation an explanation of how they conducted their analyses (move appropriate part from materials and method).

We agree and added more information about the statistical and experimental setup to the results sections (lines 177-179, 191-193, 199, 210-212).

Also, there is no information on the type of chemotherapy and TKI-targeted therapies in the cohorts concerned.

We agree and added the information about individual therapies to table 1 .

A reorganization will make the results better to read.

In line with a comment from above, we fitted the results part to make the results clearer and easier to follow.

Please provide a list of abbreviation, so that the reader will be easy to follow it.

We added an abbreviation list in the appendix section.

Conclusion should be improved and little extended.

We wrote the conclusions in accordance with the guidelines of Cancers (Conclusions: This section is mandatory, with one or two paragraphs to end the main text). With respect to these guidelines, we only slightly extended this part (lines 398-402).

Reviewer 2 Report

The manuscript “Novel liquid biomarker panels for a very early response capturing of NSCLC therapies in advanced stages” by Janke F et al addresses a relevant clinical question. The authors show interesting results with translational relevance in the oncological field, but some validations would be needed to give force to their findings. In particular, the authors, where it is possible, should show the correlation between the progression free survival (PFS) of the treated patients and the measurement of the selected biomarkers.

There are some other additional comments:

  • In the Introduction the authors should better describe the aim of the study.
  • In the Discussion the authors should better describe the clinical implication of their results.

Author Response

We thank the author for the valuable comments. We improved the manuscript and attached the point-by-point comments below:

In particular, the authors, where it is possible, should show the correlation between the progression free survival (PFS) of the treated patients and the measurement of the selected biomarkers.

We agree that survival analyses using the PFS of the patients might be another interesting aspect of the study. However, we believe that this is not the aim of our current study where we were interested in whether we can find a marker panel that indicate a successful or failed therapy in terms of response, before it becomes visible in radiographic tumor assessments.

In the Introduction the authors should better describe the aim of the study.

We agree and described the aim of the study more in detail (lines 115-116).      

In the Discussion the authors should better describe the clinical implication of their results.

Thank you very much for this hint. We added a part in the discussion section where we considered the clinical implication of our results more in detail (lines 289-297).

Reviewer 3 Report

First, thank you for giving me the opportunity to review this manuscript. English is not my first

language, so I am not able to correct all grammar mistakes (if present)

General comments:

This study deals with the detection of biomarkers at different time after treatment on blood samples of NSLC. The authors have analyzed two cohorts of patients treated either by chemotherapy either by TKI. This paper shows interesting information, although the authors have tested a very limited number of patients.

I have some minor comments:

  • Some minor typos errors are present trough the manuscript: ie in abstract section “day +14).Our…”, there is space missing after the “.” . “The detection” where “The” is in bold…etc

I have some major comments:

  • The studied population is not homogeneously tested, with only 1 sample at day 1 for patients treated by chemotherapy and 2 blood samples at day 7 and day 14.
  • Figure 4. Predictive value of single markers at the three time points and correlation analyses. -> I would like to see the results for all other time points. Was there a difference before treatment and at +1 day?
  • How the patients included were selected? Were there all consecutive patients with NSCLC or selected by physicians?
  • Why not restrain the study on adenocarcinoma? Although lung adenocarcinoma is the most frequent lung carcinoma, the proportion of adenocarcinoma, squamous cell carcinoma and NOS are not representative of the proportion encountered in real life. Due to the hight proportion of lung adenocarcinoma, I think the information brought by this manuscript is important for adenocarcinoma and not for NSLC. I think the data for squamous cell carcinoma and carcinoma NOS should not be included in the analysis and the title modified.

I did not detect plagiarism. I do not have Ethic concern. I did not detect image manipulation

My suggestion is: Major revision

Author Response

We thank the reviewer for the constructive comments. We adapted and improved the manuscript. Please find our point-by-point comments below:

The studied population is not homogeneously tested, with only 1 sample at day 1 for patients treated by chemotherapy and 2 blood samples at day 7 and day 14.

The study design is a result of our literature search. There are hints that a classical chemotherapy can affect the tumor faster than a TKI treatment, which is classically administered orally compared to administration of intravenous chemotherapy (see also introduction part). In contrast to intravenous chemotherapy which diffuses into the organs and thereby into tumor tissue very shortly after i.v. administration, oral TKIs need between 7 to 14 days to reach a steady state in the patient’s system. Thus, we examined patients under TKI with a different chronology as compared to the patients undergoing chemotherapy. We added more information on this to the MM section (lines 310-324).

Nevertheless, since both cohorts are analyzed separately, the difference in collection of blood samples should have no influence on our findings.

Figure 4. Predictive value of single markers at the three time points and correlation analyses. -> I would like to see the results for all other time points. Was there a difference before treatment and at +1 day?

We apologize but obviously, the description behind these results was not written clear enough. We correlated the values of the biomarkers measured at the time points +1, +7 and +14 with the tumor load change that was assessed as size change between baseline (day -1) and first follow-up CT scan.

-> I would like to see the results for all other time points.

We show the results for all time points.

Was there a difference before treatment and at +1 day?

We do not really know what the reviewer means with this question. We do not show the baseline here since no therapy is applied at this time and it would not make sense to correlate the baseline values to the tumor load change. In figure 3, we correlated the biomarkers to initial tumor load, if this is the question here. 

We made changes in the material and method section (lines 310-324) as well as in result part (lines 122-130, 177-179, 191-193, 199, 210-212) to make the experimental and statistical setup clearer. We hope the result part is now easier to follow.

How the patients included were selected? Were there all consecutive patients with NSCLC or selected by physicians?

Our biomarker study was a part of a larger clinical study in which the patients were selected by clinicians requiring MRI and a measurable tumor size of at least 2 cm. All patients suffered from NSCLC and were selected by recommended therapy concept (TKI or chemotherapy) according to the current guidelines. We added more information about the cohort in the material and methods section to follow your comments (lines 310-324).

Why not restrain the study on adenocarcinoma? Although lung adenocarcinoma is the most frequent lung carcinoma, the proportion of adenocarcinoma, squamous cell carcinoma and NOS are not representative of the proportion encountered in real life. Due to the hight proportion of lung adenocarcinoma, I think the information brought by this manuscript is important for adenocarcinoma and not for NSLC. I think the data for squamous cell carcinoma and carcinoma NOS should not be included in the analysis and the title modified.

We agree that adenocarcinoma is the most prominent histology in our cohort. This a result of clinical selection of the patients. Patients with adenocarcinoma are routinely analyzed for molecular alterations in contrast to patients with squamous cell carcinoma. Large TCGA studies revealed that adenocarcinoma often show targetable driver mutations while squamous cell carcinoma mainly contain deletion mutations. The presence or absence of a driver mutation was the major criterion for the separation of the patients in two groups. Beside the adenocarcinoma, the two other histologies (large cell, squamous cell) are rarely tested for molecular alterations and therefore not frequent in our whole cohort. Nevertheless, molecular testing is performed more and more also for non-adeno patients. Therefore, we believe that our panels can be valuable for all NSCLC histologies. A validation with a larger cohort is essential and warranted.

Round 2

Reviewer 1 Report

The authors have addressed my comments. I support publication of this manuscript. 

Reviewer 3 Report

The authors responded to my comments. Ido not have any more comment.

I suggest to accept this manuscript.